# Proteome Analysis of Aflibercept Intervention in Experimental Central Retinal Vein Occlusion

**DOI:** 10.3390/molecules27113360

**Published:** 2022-05-24

**Authors:** Lasse Jørgensen Cehofski, Anders Kruse, Alexander Nørgaard Alsing, Benn Falch Sejergaard, Jonas Ellegaard Nielsen, Anders Schlosser, Grith Lykke Sorensen, Jakob Grauslund, Bent Honoré, Henrik Vorum

**Affiliations:** 1Department of Ophthalmology, Odense University Hospital, 5000 Odense C, Denmark; jakob.grauslund@rsyd.dk; 2Department of Ophthalmology, Aalborg University Hospital, 9000 Aalborg, Denmark; anders.kruse@rn.dk (A.K.); noergaard.alsing@rn.dk (A.N.A.); b.sejergaard@rn.dk (B.F.S.); henrik.vorum@rn.dk (H.V.); 3Department of Clinical Research, University of Southern Denmark, 5000 Odense C, Denmark; 4Department of Biomedical Research Laboratory, Aalborg University Hospital, 9000 Aalborg, Denmark; 5Department of Clinical Biochemistry, Aalborg University Hospital, 9000 Aalborg, Denmark; jonas.ellegaard.nielsen@rsyd.dk; 6Department of Clinical Medicine, Aalborg University, 9000 Aalborg, Denmark; bh@biomed.au.dk; 7Department of Cancer and Inflammation Research, University of Southern Denmark, 5000 Odense C, Denmark; aschlosser@health.sdu.dk (A.S.); glsorensen@health.sdu.dk (G.L.S.); 8Department of Biomedicine, Aarhus University, 8000 Aarhus C, Denmark

**Keywords:** retina, retinal vein occlusion, mass spectrometry, proteomics, proteome, aflibercept, vascular endothelial growth factor

## Abstract

Aflibercept is a frequently used inhibitor of vascular endothelial growth factor (VEGF) in the treatment of macular edema following central retinal vein occlusion (CRVO). Retinal proteome changes following aflibercept intervention in CRVO remain largely unstudied. Studying proteomic changes of aflibercept intervention may generate a better understanding of mechanisms of action and uncover aspects related to the safety profile. In 10 Danish Landrace pigs, CRVO was induced in both eyes with an argon laser. Right eyes were treated with intravitreal aflibercept while left control eyes received isotonic saline water. Retinal samples were collected 15 days after induced CRVO. Proteomic analysis by tandem mass tag-based mass spectrometry identified a total of 21 proteins that were changed in content following aflibercept intervention. In retinas treated with aflibercept, high levels of aflibercept components were reached, including the VEGF receptor-1 and VEGF receptor-2 domains. Fold changes in the additional proteins ranged between 0.70 and 1.19. Aflibercept intervention resulted in a downregulation of pigment epithelium-derived factor (PEDF) (fold change = 0.84) and endoplasmin (fold change = 0.91). The changes were slight and could thereby not be confirmed with less precise immunohistochemistry and Western blotting. Our data suggest that aflibercept had a narrow mechanism of action in the CRVO model. This may be an important observation in cases when macular edema secondary to CRVO is resistant to aflibercept intervention.

## 1. Introduction

Central retinal vein occlusion (CRVO) is a visually disabling condition caused by a thrombus of the central retinal vein, which is the major outflow vessel of the eye [1,2]. Occlusion of the central retinal vein results in increased resistance to blood flow in retinal arterioles. The reduced blood flow causes closure of retinal capillaries and small arterioles, resulting in retinal hypoxia, which drives an increased production of vascular endothelial growth factor A (VEGF-A) and a complex inflammatory response mediated by interleukin (IL)-6, IL-8 and monocyte chemotactic protein-1 [3,4]. VEGF-A and the inflammatory response increase vascular permeability resulting in macular edema, which is the most common cause of vision loss in CRVO [4].

Macular edema secondary to CRVO is effectively treated with intravitreal injections of anti-VEGF agents including bevacizumab, ranibizumab and aflibercept, which reduce retinal vascular permeability and cause absorption of the macular edema [5].

Aflibercept is a frequently used inhibitor of VEGF and has a well-documented efficacy in the treatment of macular edema secondary to CRVO [6,7]. It consists of a constant Fc domain of human immunoglobulin G1 fused with the second immunoglobulin domain of VEGF receptor-1 (VEGFR-1) and the third immunoglobulin domain of VEGF receptor-2 (VEGFR-2) [8,9]. Although aflibercept treatment has become the standard of care, retinal large-scale protein changes following aflibercept intervention in CRVO remain largely unstudied [10,11].

The overall objective of proteomic studies is to identify and quantify the entire set of proteins in a given cell, tissue or biofluid to provide insights into biological processes in the disease or intervention under study [11,12,13]. Studying retinal proteome changes following aflibercept intervention in CRVO may bring important therapeutic insights into the mechanism of action of aflibercept. Furthermore, studying retinal proteome changes with proteomic techniques may bring insights into the safety profile of aflibercept. Elucidating the retinal proteome in CRVO following aflibercept intervention may also provide a potential for discoveries that can lead to the improvement of existing therapies [10,11].

Retinal tissue exposed to CRVO is generally only available from animal models. In the study presented, aflibercept was tested in a well-established porcine model of laser-induced CRVO [14], which is suited for expressional studies due to its non-invasive nature. Advanced proteomic techniques were used to study large-scale retinal protein changes following aflibercept intervention in the CRVO model.

## 2. Results

### 2.1. Evaluation of Experimental CRVO Model

In porcine eyes with CRVO, flame-shaped hemorrhages and venous dilation were observed upstream of the site of occlusion within 30 min after inducing CRVO (Figure 1A–C). Fluorescein angiography was performed three days after CRVO to confirm that CRVO was successfully induced. Fluorescein angiography of eyes with CRVO showed delayed filling of retinal branch veins, retinal capillary non-perfusion and leakage around retinal veins (Figure 2A–D). Retinal capillary non-perfusion was observed in all retinal quadrants.

Data output from MaxQuant are available in the Appendix A.

We first tested the reproducibility of proteome changes in the CRVO model in seven animals (five animals were used for mass spectrometry and Western blotting, and two animals were used for immunohistochemistry). With tandem mass tag (TMT)-based mass spectrometry and Western blotting we compared CRVO (*n* = 5) induced right eyes with the left control eyes (*n* = 5), which received laser without inducing occlusion. Overall, retinal proteome changes in the CRVO model (Appendix A) were consistent with previous findings in the model [14], including an upregulation of fibronectin (fold change = 13.50, *p* = 0.0016) and galectin-3 (fold change = 5.31; *p* = 0.0016) as well as a downregulation of neurofilament light polypeptide (*p* = 0.036; fold change = 0.31). Samples from one animal were excluded from the dataset as the samples were not successfully labeled with the TMT kit. Western blotting confirmed the increased content of galectin-3 in CRVO (*n* = 5) vs. control (*n* = 5) (Figure 3A). The regulation of fibronectin, galectin-3 and neurofilament light polypeptide was confirmed by immunohistochemistry comparing the CRVO (*n* = 2) vs. control (*n* = 2) (Figure 3B–G). Additional immunohistochemistry is available in the Appendix A.

### 2.2. Retinal Proteome Changes Following Aflibercept Intervention in Experimental CRVO

Retinal proteome changes following aflibercept intervention were studied in eight Danish Landrace pigs. CRVO was induced in both eyes. Right eyes were treated with aflibercept (*n* = 8) while left control eyes (*n* = 8) were treated with saline water (NaCl). A total of 3559 proteins were successfully assigned and quantified from the retinal samples (Appendix A). A total of 21 proteins were significantly changed in content following aflibercept intervention in the CRVO model (Table 1).

High contents of aflibercept VEGF receptor domains and the aflibercept fusion protein were observed in retinas treated with aflibercept, including the VEGFR-1 immunoglobulin domain (fold change = 46.4; *p* = 2.45 × 10^−6^), the VEGFR-2 immunoglobulin domain (fold change = 5.07; *p* = 1.95 × 10^−8^) and the fusion protein Ig gamma-1 chain C region (fold change 18.90; *p* = 3.27 × 10^−11^). Changes in all other proteins were small with fold changes ranging between 0.70 and 1.19 (Figure 4) (Table 1). Aflibercept intervention in the CRVO model resulted in an upregulation of A-kinase anchor protein 8 (AKAP8) and a downregulation of endoplasmin (fold change = 0.91; *p* = 0.041), and pigment epithelium-derived factor (PEDF) (fold change = 0.84; *p* = 0.020). However, the slight changes in endoplasmin and PEDF detected by mass spectrometry were too small to be confirmed with the less precise techniques of immunohistochemistry (Figure 5) and Western blotting (Figure 6). Immunohistochemistry showed by eye a similar staining pattern of endoplasmin and PEDF regardless of aflibercept intervention (Figure 5). Western blotting showed a slight downregulation of PEDF and endoplasmin following aflibercept intervention (Figure 6A–D). However, the differences were not statistically significant as the standard deviations of Western blotting data were higher than observed with mass spectrometry data (Figure 6E–G).

Aflibercept intervention in CRVO also resulted in downregulation of 40S ribosomal protein S18 (fold change = 0.76; *p* = 0.030) and mitochondrial 28S ribosomal protein S7 (fold change = 0.70; *p* = 0.026). Fold changes in the additional proteins ranged between 0.82 and 1.19 (Table 1).

When proteins were listed according to abundance regardless of the p-value, the largest changes were observed for aflibercept domains (Table 2). The protein with the most pronounced downregulation was alpha-crystallin A chain (CRYAA), which was close to being significantly regulated (*p* = 0.07) (Table 2).

## 3. Discussion

### 3.1. Evaluation of Experimental CRVO Model

Angiography confirmed that the model had angiographic similarities with CRVO in humans with the occlusion emerging from the optic nerve head, generating retinal capillary non-perfusion in all quadrants of the retina. Recanalization of CRVO was not observed in any of the animals. Proteome changes in the CRVO model were similar to a previous study of the model [14] confirming a high reproducibility at the molecular level in the model.

### 3.2. Aflibercept Intervention in CRVO

Results from the proteomic analysis indicated that aflibercept did not regulate multiple signaling pathways in the CRVO model. This is an important observation in terms of the safety profile of aflibercept. Thus, our data suggest that aflibercept did not regulate pathways, which could have negative side effects on the retina. Very high levels of aflibercept components were reached after 15 days of treatment, indicating high retinal concentrations of the compound in the CRVO model.

Observed protein changes following aflibercept intervention were very small. Two proteins were selected for further validation, endoplasmin and PEDF, but their regulation was not confirmed with immunohistochemistry and Western blotting. A downregulation of CRYAA was observed following aflibercept intervention, but the change was not statistically significant (*p* = 0.07). Knock-out of CRYAA has been reported to inhibit ocular neovascularization in a murine model of oxygen-induced retinopathy [15], but more studies will be needed to establish a relation between CRYAA regulation and aflibercept treatment. Our data indicate that aflibercept had a narrow mechanism of action in CRVO. In a clinical setting, this may an important observation in cases when macular edema secondary to CRVO is resistant to aflibercept intervention. In cases when VEGF is not a major driving force in macular edema secondary to CRVO, aflibercept may have a limited effect due to a narrow mechanism of action. Our study has important implications for patients receiving aflibercept treatment. Our data suggest that aflibercept does not regulate a multitude of proteins or pathways that may be unwanted or result in side effects.

Proteome analysis of the retina implies a number of limitations that may affect the outcome of the proteomic analysis [12]. We have previously shown that proteome changes in the retina often occur in specific retinal layers or cell types [16]. As a consequence, observed proteome changes may be moderate when the entire retina is collected for proteomic analysis instead of isolating specific cells or cell layers. Due to the multi-layered structure of the retina, proteome studies of retinal tissue are best supported by immunohistochemistry. More than 3000 retinal proteins were successfully assigned. However, the multilayered structure of the retina adds further complexity to the proteomic analysis in terms of detection of low abundance proteins, as protein abundances stretch over multiple orders of magnitude [12].

## 4. Materials and Methods

### 4.1. Animal Preparation

The study was approved by the Danish Animal Experiments Inspectorate, permission no. 2019-15-0201-01651. Danish Landrace pigs were housed under a 12 h light/dark cycle, and general anesthesia, topical anesthesia with eye drops and dilation of the pupils were performed as previously described [17].

### 4.2. Experimental CRVO

The study used an experimental model of CRVO as described previously [14]. We first verified the reproducibility of the CRVO model at the molecular level in seven Danish Landrace pigs (five animals were used for mass spectrometry and Western blotting, while two animals were used for immunohistochemistry). In these animals, CRVO was induced in the right eyes, while left control eyes received laser without inducing CRVO. In the right eyes, CRVO was induced close to the optic nerve head with a standard argon laser (532 nm) given by indirect ophthalmoscopy using a 20D lens. The laser energy was set to 400 mW with an exposure time of 550 ms. A total of 30–40 laser applications were used per occlusion. By applying the laser directly on retinal veins close to the optic nerve head, thrombotic material was directed towards the optic nerve head and the lamina cribrosa. Experimental CRVO was considered successful when stagnation of venous blood and development of flame-shaped hemorrhages were observed by ophthalmoscopy. In the left control eyes, a laser control without occlusion was created by giving the same amount of laser applications and energy level at the edge of the optic nerve head, but without inducing occlusion. CRVO was confirmed with fluorescein angiography, and the eyes were dissected 15 days after induced CRVO and saved for mass spectrometry and immunohistochemistry. Fifteen days after induced CRVO, the eyes were enucleated. The eyes were dissected on ice under a microscope. The anterior segment was removed. The vitreous body was aspired into a 5 mL syringe. In eyes intended for proteomic analysis, the neurosensory retina was peeled from the RPE/choroid complex with tweezers and stored at −80 °C. In eyes intended for immunohistochemistry, complexes consisting of neurosensory retina, RPE/choroid complex and sclera were excised for immunohistochemistry. The animals were euthanized immediately after enucleation.

To test aflibercept intervention in CRVO, another 10 Danish Landrace pigs were used. In these animals CRVO was induced in both eyes as described above. An intravitreal injection of 0.05 mL aflibercept 40 mg/mL (Bayer, Leverkusen, Germany) was given in the right eyes, while left eyes received an injection of 0.05 mL sodium chloride 9 mg/mL (NaCl) (B. Braun, Frederiksberg, Denmark). Following the injections, chloramphenicol ointment 1% (Takeda Pharma A/S, Taastrup, Denmark) was applied in both eyes. Fluorescein angiography was performed three days after CRVO to confirm that CRVO was induced successfully. The eyes were dissected 15 days after CRVO as described above.

### 4.3. Sample Preparation for Mass Spectrometry

The reproducibility of proteome changes in the CRVO model was verified by comparing CRVO (*n* = 5) vs. laser control (*n* = 5) with tandem mass tag (TMT)-based mass spectrometry in a separate analytical run. Eyes from eight animals were used to compare the protein profile of CRVO + aflibercept (*n* = 8) vs. CRVO + NaCl (*n* = 8) with proteomic analysis by tandem mass tag (TMT)-based mass spectrometry.

Isobaric labeling was performed with a 10 plex TMT kit from Thermo Scientific (Waltham, MA, USA). Sample preparation for TMT-based mass spectrometry was performed as previously described [14,16] with some modifications. For the experiment consisting of 16 samples, a standard was prepared by mixing equal amounts from each sample. Two groups of 10 samples, 8 experimental samples together with 2 standards, were labelled with the 10 plex kit. The standards were used for normalization of data. TMT labeling and high pH reversed phase peptide fractionation were performed as described in a previous article [18]. Then, 1 µg of fractions 2–8 was analyzed.

### 4.4. Quantification with Tandem Mass Tag-Based Mass Spectrometry

One microgram of each fraction was loaded for each run onto a Dionex UltiMateTM 3000 RSLC nanosystem coupled to an Orbitrap Fusion mass spectrometer (Thermo Scientific, Waltham, MA, USA) equipped with an EasySpray^TM^ ion. Liquid chromatography and mass spectrometry with TMT synchronous precursor selection MS^3^ mode was performed as follows. The labeled samples were loaded onto the trapping column (5 mm × 300 µm, C18 PepMap100, 5 µm, 100 Å, Thermo Scientific, Waltham, MA, USA) with the flow setting of 30 µL per min. The nanoflow was 300 nL per min for the separation of peptides on the analytical column (500 mm × 75 µm PepMap RSLC, C18, 2 µm, 100 Å, Thermo Scientific, Waltham, MA, USA). The applied buffers were buffer A (99.9% water and 0.1% formic acid) and buffer B (99.9% acetonitrile and 0.1% formic acid). The gradient was performed over 213 min with a gradient of buffer B ranging from 2% to 80%.

The mass spectrometer was operated in the TMT SPS MS^3^ mode with full Orbitrap scans in the mass range of 350–1500 *m*/*z* obtained at a resolution of 120,000 with an AGC target of 2 × 10^−5^ and a maximum injection time of 50 ms. The mass spectrometer was set to trigger MS^2^ acquisitions in each cycle using the linear ion trap with a CID collision energy at 35% and an AGC target of 2 × 10^4^ with a maximal injection time of 75 ms. Precursor ions, in the mass range of 400–1200 *m*/*z*, were isolated in the quadrupole set with an isolation window of 1.2 *m*/*z*. Up to five reporter ions were detected in MS^3^ with synchronous precursor selection performed in the Orbitrap in the mass range of 100–500 *m*/*z* with the HCD collision energy set to 65%, obtained at a resolution of 50,000 and an AGC target of 3 × 10^4^ and a maximum injection time of 110 ms. A dynamic exclusion of 6 s was applied.

With MaxQuant software version 1.6.6.0, accessed on 18 September 2019 and on 28 January 2021 (Max Planck Institute of Biochemistry, Martinsried, Germany; https://maxquant.net/maxquant/), raw data files were searched against the Uniprot *Sus scrofa* and *Homo sapiens* databases using match between runs and with settings described in a previous work [16]. The data output from MaxQuant is available in the Appendix A.

### 4.5. Filtration of Proteins and Statistics

The reproducibility of the CRVO model was assessed through statistical analysis in Perseus version 1.6.6.0 (Max Planck Institute of Biochemistry, Martinsried, Germany; https://maxquant.net/perseus/ accessed on 18 September 2019) as previously described [14], with the only exceptions that the number of randomizations was set to 250, the S_0_ parameter was set to 0.1 and a false discovery rate of 0.05 was applied.

To compare CRVO + aflibercept vs. CRVO + NaCl in Perseus software version 1.6.14.0 (Max Planck Institute of Biochemistry, Martinsried, Germany; https://maxquant.net/perseus/ accessed on 18 September 2019), poorly assigned proteins were removed in Perseus as described in a previous article [14]. Proteins were required to be successfully assigned and quantified in 100% of the samples in each group. Quantitative values were log_2_ transformed, and technical replicates were averaged. At least two unique peptides were required for successful identification. A Student’s *t*-test was performed in Perseus to compare CRVO + aflibercept vs. CRVO + NaCl. Proteins were considered significantly changed in content if *p* < 0.05.

### 4.6. Immunohistochemistry

Eyes from two animals were used to verify the reproducibility of the CRVO model at the molecular level comparing CRVO (*n* = 2) vs. laser control (*n* = 2). Eyes from two animals were used to compare CRVO + aflibercept (*n* = 2) vs. CRVO + NaCl (*n* = 2).

Complexes consisting of retina, choroid and sclera were fixated in formalin for 24 h. The formalin solution was removed. The tissue was then stored in a PBS solution at 4 °C until further use. Then, 4 µm thick sections were cut from NBF-fixed paraffin-embedded tissue blocks. Sections were mounted on FLEX IHC Slides (Dako; Glostrup, Denmark), dried at 60 °C, dewaxed and rehydrated through a graded ethanol series, and they were subsequently washed in 0.05 M Tris-buffered saline (TBS; Fagron Nordic A/S; Copenhagen, Denmark). Endogenous biotin reactivity was blocked with 1.5% hydrogen peroxide. Optimal epitope retrieval was performed using microwave heating for 11 min at full power (900 W), followed by 15 min at 400 W in 10 mM Tris (Fagron Nordic A/S; Copenhagen, Denmark) with 0.5 mM ethylene glycol-bis(β-aminoethyl ether)-*N,N,N*′,*N*′-tetraacetic acid (EGTA; Fagron Nordic A/S) at pH 9.0 (TEG buffer). After heating, slides remained in buffer for 15 min. Primary antibodies used for immunohistochemistry included a polyclonal rabbit anti-fibronectin antibody (A0245, Dako, Glostrup, Denmark) 1:1000, a primary polyclonal rabbit anti-galectin-3 antibody (MBS3211803, MyBioSource, San Diego, CA, USA) 1:200, a monoclonal mouse anti-neurofilament light polypeptide (M0762, Dako, Glostrup, Denmark) 1:100, a rat monoclonal anti-endoplasmin antibody (MBS439463, MyBioSource, San Diego, CA, USA) and a rabbit polyclonal anti-pigment epithelium derived factor (PEDF) antibody (MBS2027143, MyBioSource, San Diego, CA, USA).

Sections were then incubated for 60 min with antibodies diluted in TNT Antibody Diluent (Dako, Glostrup, Denmark A/S). Visualization of the antigen–antibody complex was carried out with the PowerVision+ (Leica, Copenhagen, Denmark A/S) detections system according to the manufacturer’s manual. DAB was used as a chromogen (K3468, Dako, Glostrup, Denmark). Immunostaining was followed by brief nuclear counterstaining in Mayer’s hematoxylin (Fagron Nordic A/S, Copenhagen, Denmark). Finally, slides were washed, dehydrated and coverslipped using a Tissue-Tek Film coverslipper (Sakura Finetek; Alphen aan den Rijn, The Netherlands). Histology slides were scanned at 40× magnification using a NanoZoomer-XR (Hamamatsu Photonics; Hamamatsu City, Japan), and image acquisition was obtained using NDP.view2 software (NanoZoomer Digital Pathology; Hamamatsu Photonics; Hamamatsu City, Japan).

### 4.7. Western Blotting

The reproducibility of the CRVO model was verified by comparing galectin-3 levels in CRVO (*n* = 5) vs. controls (*n* = 5) using a primary polyclonal rabbit anti-galectin-3 antibody 1:100 (MBS3211803, MyBiosource, San Diego, CA, USA). Western blotting was used to quantify galectin-3 using beta-tubulin as a housekeeping protein loading control as previously described [14]. A Student’s *t*-test was performed on logarithmized densitometric data.

Western blotting was used to compare CRVO + aflibercept (*n* = 8) vs. CRVO + NaCl (*n* = 8) and was performed as follows. Protein concentrations were determined using the DC™ Protein Assay Kit II (Bio-Rad, Copenhagen, Denmark) according to the manufacturer’s instructions. First, 50 µg of each sample was mixed with 6 × SDS-PAGE buffer (containing 40% (*v*/*v*) glycerol, 8% (*w*/*v*) SDS, 25% (*v*/*v*) 125 mM Tris-base (pH 8.6) and 0.002% bromphenol blue) and added dithiothreitol (DTT) to a concentration of 40 mM. The samples were heated to 95 °C for 3 min and finally alkylated using 15 × 1.4 M iodoacetamide IAA. Protein lysates were separated on polyacrylamide pre-casted gradient gels (Bolt 4–12% Bis-Tris Plus, Invitrogen NW04125, Thermo Fisher, Waltham, MA, USA) with the NuPage electrophoresis system (Thermo Fisher, Waltham, MA, USA) using the NuPage MOPS SDS running buffer Invitrogen NP0001 (Thermo Fisher, Waltham, MA, USA) and blotted onto a polyvinylidene difluoride membrane (PVDF) membrane (Amersham Hybond P, 10600021). The pre-stained Precision Plus Protein™ Kaleidoscope™ Protein Standard (Bio-Rad, Copenhagen, Denmark, 1610375) was used.

Proteins were blotted from the gels to the membranes using the NuPage system with the XCell II™ Blot Module (Thermo Fisher, Waltham, MA, USA) and transfer buffer with 25 mM Tris-base, 195 mM Glycine, 10% SDS and 96% ethanol. After activation and protein transfer, the membranes were blocked for 1 h at room temperature in ROTIBloc A151.1 (Carl Roth, Karlsruhe, Germany) and then incubated with primary antibodies diluted in ROTIBlock overnight.

The primary antibodies used were a rat 1 µg/mL monoclonal anti-endoplasmin antibody (MBS439463, MyBioSource, San Diego, CA, USA) and a rabbit polyclonal 2 µg/mL anti-pigment epithelium-derived factor (PEDF) antibody (MBS2027143, MyBioSource, San Diego, CA, USA). A monoclonal anti-GAPDH antibody (Sc-32233, Santa Cruz, Dallas, TX, USA) was also used. On the following day, the membranes were washed in TBST (20 mM Tris, 150 mM NaCl and 0.1% Tween 20 adjusted to pH 7.4) and incubated for 1 h with corresponding secondary antibodies diluted in TBST including 1:20.000 Goat-anti-Rabbit-HRP P0448 (Dako, Glostrup, Denmark), 1:10,000 Goat anti-Rat-HRP Invitrogen 31470 (Thermo Fisher, Waltham, MA, USA) and 1:20,000 Rabbit Anti Mouse-HRP DAKO P0260 (Dako, Glostrup, Denmark). Membranes were then washed in TBST and developed using the SuperSignal Chemiluminescent S kit (Thermo Fisher, Waltham, MA, USA) according to the manufacturer’s instructions followed by using the BioRad Chemidoc developer machine (Bio-Rad, Copenhagen, Denmark). ImageJ 1.8.0_172 was used to quantify the intensity of the bands. Statistical analysis was performed with a Student’s *t*-test on densitometric data. Power calculations related to Western blotting were performed with STATA 16.0 (StataCorp, College Station, TX, USA) using the power calculation for the *t*-test comparing two independent means.

## 5. Conclusions

Proteomic analysis identified a total of 21 proteins that changed in content following aflibercept intervention in experimental CRVO. High retinal levels of aflibercept components were observed 15 days after aflibercept intervention. Thus, high retinal aflibercept concentrations are reached 15 days after aflibercept intervention in CRVO. Except from high levels of aflibercept components, the protein changes observed with aflibercept treatment were very small. The regulation of selected proteins was too slight be confirmed with immunohistochemistry and Western blotting. Our data suggest that aflibercept had a narrow mechanism of action in the CRVO model. In a clinical setting, this may an important observation in cases when macular edema secondary to CRVO is resistant to aflibercept intervention. From a safety perspective, it is an important finding that aflibercept treatment did not result in major regulations of multiple signaling pathways.

## Figures and Tables

**Figure 1 molecules-27-03360-f001:**
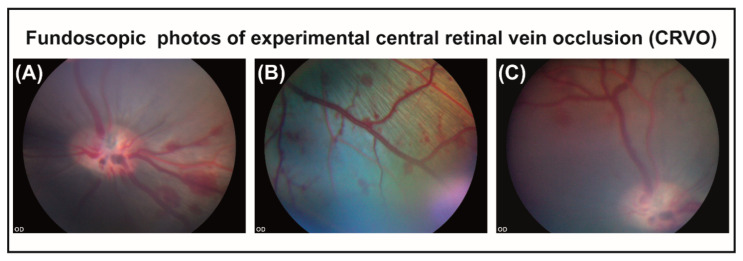
(**A**) Flame-shaped hemorrhages appearing in the retina within 30 min after CRVO. (**B**) Flame-shaped hemorrhages appearing upstream of the site of occlusion. (**C**) Dilated vein observed upstream of the site of occlusion.

**Figure 2 molecules-27-03360-f002:**
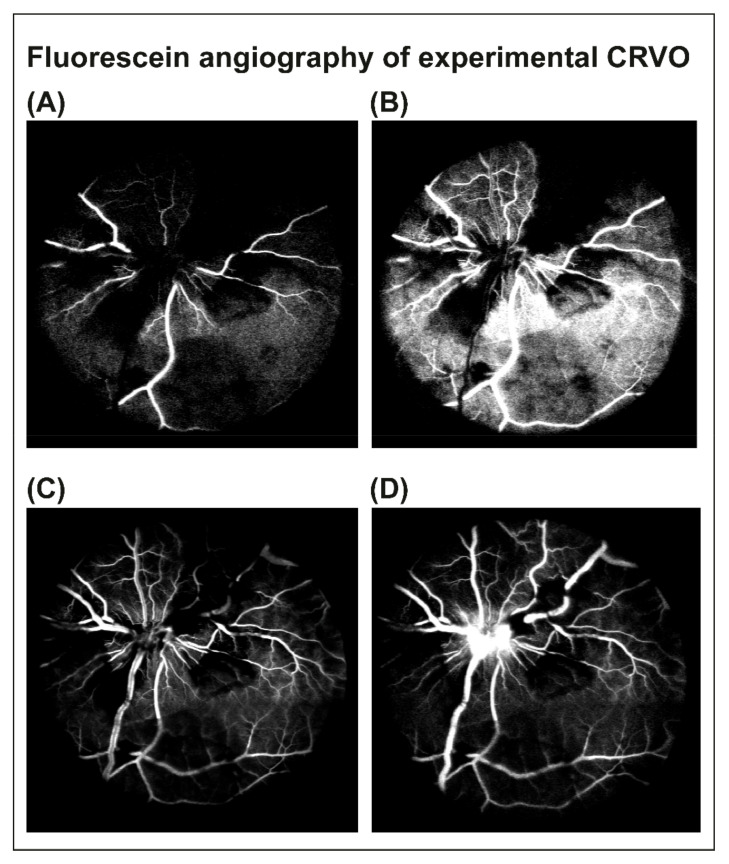
Fluorescein angiography was performed 3 days after CRVO to confirm that the occlusion was successfully induced. (**A**–**C**) Early phases of fluorescein angiography of experimental CRVO at 15–24 s. Angiography revealed tortuous retinal veins, delayed venous filling and retinal capillary non-perfusion following CRVO. (**D**) Angiography at 1 min and 26 s. Venous dilation, retinal capillary non-perfusion and leakage of fluorescein were observed.

**Figure 3 molecules-27-03360-f003:**
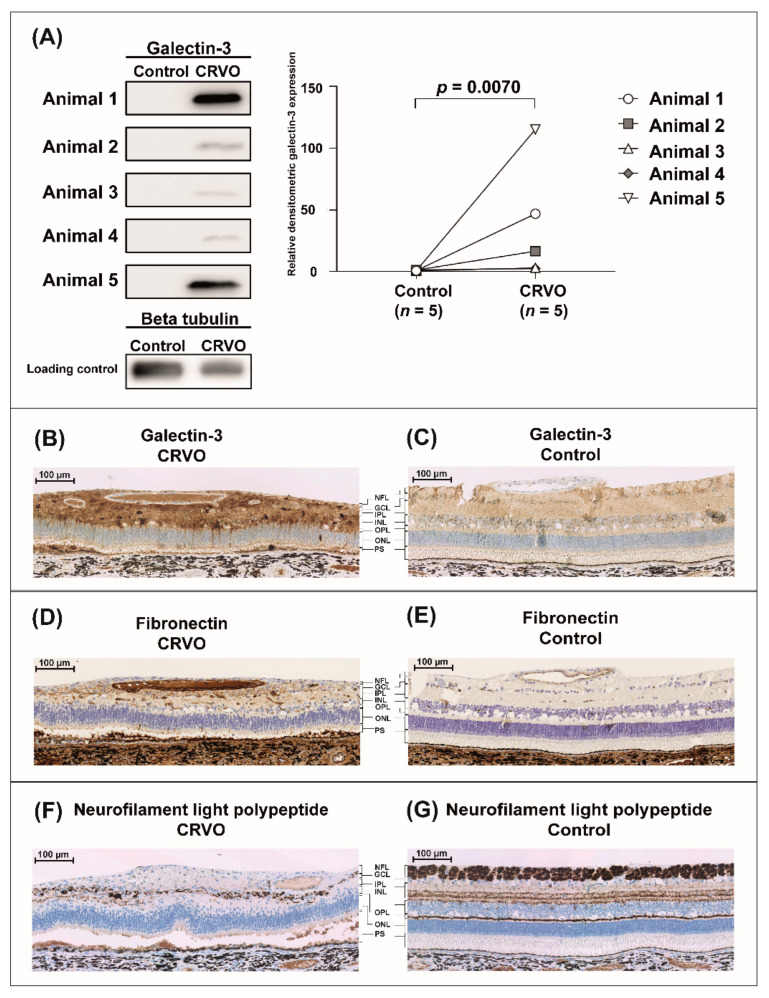
Reproducibility of the CRVO model at the molecular level. The reproducibility of the CRVO model was tested by confirming a number of key proteins that were previously found to be regulated in the model, including galectin-3, fibronectin and neurofilament light polypeptide. Laser-induced CRVO was compared to control eyes that received laser without inducing occlusion. Representative immunohistochemistry is provided. Additional immunochemistry from an additional animal is provided in the Appendix A. (**A**) Western blotting confirmed an upregulation of galectin-3 in the CRVO model. (**B**,**C**) Immunohistochemistry confirmed an upregulation of retinal galectin-3 in Müller glial cells in CRVO. (**D**,**E**) Immunohistochemistry confirmed an upregulation of fibronectin in the endothelium of retinal vessels and in proximity to the vessels. (**F**,**G**) Immunohistochemistry confirmed a downregulation of neurofilament light polypeptide indicating axonal loss in CRVO. Scale bar = 100 µm. Reaction color: brown. Abbreviations: NFL: nerve fiber layer; GCL: ganglion cell layer; IPL: inner plexiform layer; INL: inner nuclear layer; OPL: outer plexiform layer; ONL: outer nuclear layer; PS: photoreceptor segments.

**Figure 4 molecules-27-03360-f004:**
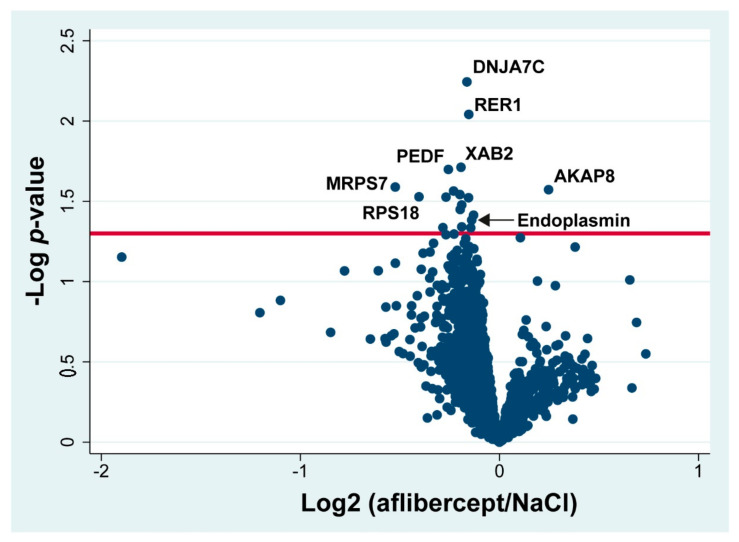
Volcano plot. Log_2_ of the ratio aflibercept/NaCl is plotted on the x-axis. On the y-axis, −log *p*-value refers to the logarithmized *p*-value from the *t*-test used to test if a protein was significantly changed. Statistically significantly changed proteins are located above the horizontal line, which denotes a significance level of 0.05. Components of aflibercept are not included in the volcano plot. PEDF: pigment epithelium-derived factor. DNAJ7C: DnaJ homolog subfamily C member 7. AKAP8: A-kinase anchor protein 8. XAB2: Pre-mRNA-splicing factor SYF1. MRPS7: 28S ribosomal protein S7, mitochondrial. RPS18: 40S ribosomal protein S18. RER1: Protein RER1.

**Figure 5 molecules-27-03360-f005:**
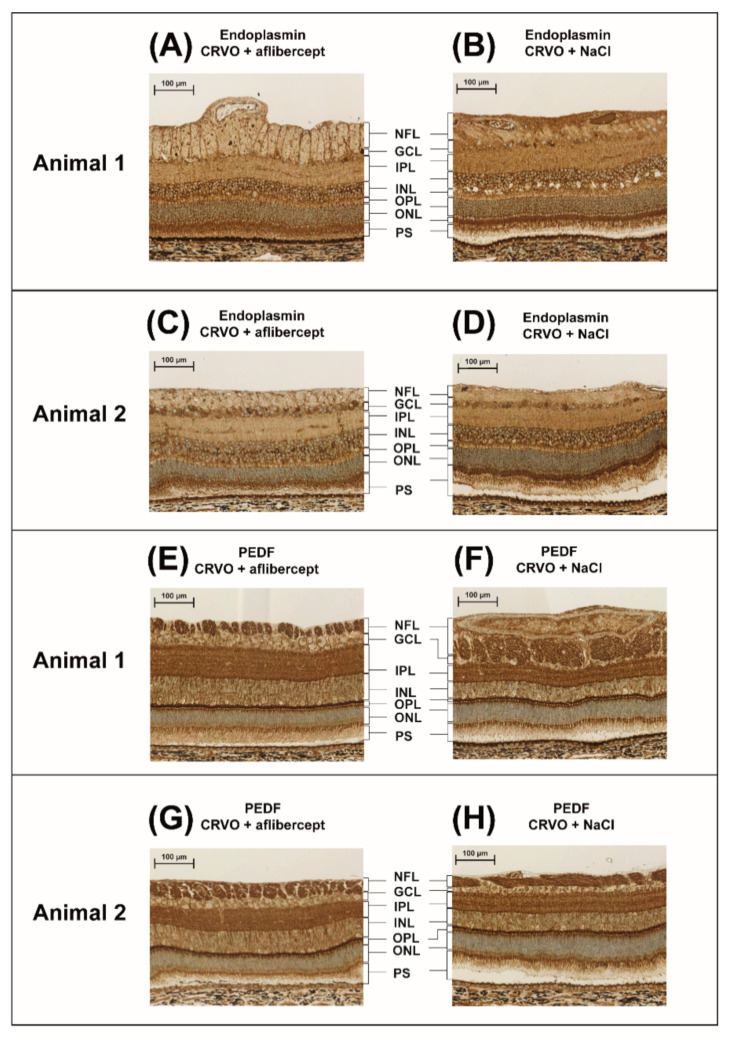
Immunohistochemistry comparing CRVO + aflibercept vs. CRVO + NaCl. Proteome analysis showed a small downregulation of endoplasmin and PEDF, which was further explored with immunohistochemistry. Immunohistochemistry showed similar staining patterns of (**A**–**D**) endoplasmin and (**E**–**H**) PEDF regardless of aflibercept intervention. Thus, immunohistochemistry of retinas treated with aflibercept did not show major changes in regulation of the endoplasmin or PEDF in specific retinal layers. Scale bar = 100 µm. Reaction color: brown. Abbreviations: NFL: nerve fiber layer; GCL: ganglion cell layer; IPL: inner plexiform layer; INL: inner nuclear layer; OPL: outer plexiform layer; ONL: outer nuclear layer; PS: photoreceptor segments.

**Figure 6 molecules-27-03360-f006:**
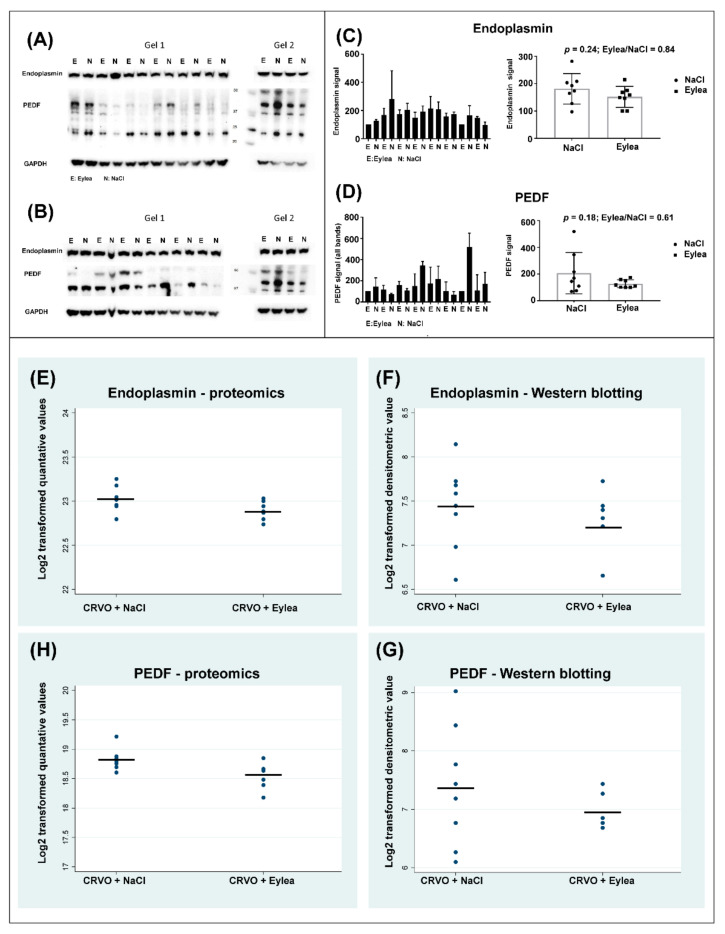
Western blots comparing CRVO + Eylea vs. CRVO + NaCl. (**A**–**B**) Western blotting was performed in two replicates. (**C**–**D**) Averages of quantitative data from the two replicates. The changes observed by mass spectrometry in endoplasmin and PEDF following aflibercept intervention were too slight to be confirmed by Western blotting. (**E**–**G**) Horizontal lines of the plots denote the means of the quantitative data. Standard deviations were larger in data obtained with Western blotting compared with quantitative data obtained through proteomic analysis. Standard deviations of endoplasmin quantification with mass spectrometry and Western blotting were 0.14 and 0.43, respectively. Standard deviations of PEDF quantification with mass spectrometry and Western blotting were 0.23 and 0.76, respectively.

**Table 1 molecules-27-03360-t001:** Proteins that significantly changed in content following aflibercept intervention.

Protein IDs	Protein Name	Gene Name	*p*-Value	Fold Change Aflibercept/NaCl
P17948	Vascular endothelial growth factor receptor 1 *	FLT1	*p* < 0.001	46.42
P0DOX5;P01857	Ig gamma-1 chain C region **	IGHG1	*p* < 0.001	18.90
P35968	Vascular endothelial growth factor receptor 2 ***	KDR	*p* < 0.001	5.07
O43823	A-kinase anchor protein 8	AKAP8	0.027	1.19
O60831	PRA1 family protein 2	PRAF2	0.039	0.91
Q29092;P14625	Endoplasmin	HSP90B1	0.041	0.91
Q9NR30-2;Q9NR30	Nucleolar RNA helicase 2	DDX21	0.046	0.90
O15258	Protein RER1	RER1	0.0091	0.90
O60518	Ran-binding protein 6	RANBP6	0.030	0.90
Q99615	DnaJ homolog subfamily C member 7	DNAJC7	0.0057	0.89
Q92522	Histone H1x	H1FX	0.046	0.88
Q9HCM4	Band 4.1-like protein 5	EPB41L5	0.033	0.88
Q9HCS7	Pre-mRNA-splicing factor SYF1	XAB2	0.019	0.87
Q53SF7	Cordon-bleu protein-like 1	COBLL1	0.036	0.87
P86791;P86790	Vacuolar fusion protein CCZ1 homolog; Vacuolar fusion protein CCZ1 homolog B	CCZ1;CCZ1B	0.029	0.87
Q9NP97	Dynein light chain roadblock-type 1	DYNLRB1	0.027	0.85
P36955	Pigment epithelium-derived factor (PEDF)	SERPINF1	0.020	0.84
Q5VZL5	Zinc finger MYM-type protein 4	ZMYM4	0.030	0.83
P83916	Chromobox protein homolog 1	CBX1	0.046	0.82
P62272;P62269	40S ribosomal protein S18	RPS18	0.030	0.76
Q9Y2R9	28S ribosomal protein S7, mitochondrial	MRPS7	0.026	0.70

* VEGFR-1 immunoglobulin domain of aflibercept. ** Ig gamma-1 chain C region of the aflibercept fusion protein. *** VEGFR-2 immunoglobulin domain of aflibercept. *p*-values refer to *p*-values from Student’s *t*-test used to identify significantly changed proteins.

**Table 2 molecules-27-03360-t002:** Proteins with fold changes >2.0 or <0.5.

Protein IDs	Protein Name	Gene Name	*p*-Value	Fold Change Aflibercept/NaCl
P17948	Vascular endothelial growth factor receptor 1 *	FLT1	*p* < 0.001	46.42
P0DOX5; P01857	Ig gamma-1 chain C region **	IGHG1	*p* < 0.001	18.9
P35968	Vascular endothelial growth factor receptor 2 ***	KDR	*p* < 0.001	5.07
P13618	ATP synthase-coupling factor 6, mitochondria	ATP5PF	0.13	0.47
Q95274; P62328	Thymosin beta-4; Hematopoietic system regulatory peptide	TMSB4X	0.16	0.43
P02475	Alpha-crystallin A chain	CRYAA	0.07	0.27

* VEGFR-1 immunoglobulin domain of aflibercept. ** Ig gamma-1 chain C region of the aflibercept fusion protein. *** VEGFR-2 immunoglobulin domain of aflibercept. *p*-values are from the Student’s *t*-test used to identify significantly changed proteins.

## Data Availability

Please refer to Appendix A.

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
