# Peer review of "Proteome Analysis of Aflibercept Intervention in Experimental Central Retinal Vein Occlusion"

_molecules, 2022, doi:10.3390/molecules27113360_

Round 1

Reviewer 1 Report

In this manuscript the authors study the changes of protein levels using tandem mass tag based mass spectrometry after the administration of the drug aflibercept in pigs eyes that were healthy or were suffering from central retinal vein occlusion. The paper is written in clear language and the methods include the important details to allow for reproducibility of the study. The introduction is on point in providing enough details to put conducted study into context, without being too meandering. The authors perform the analysis in a satisfying amount of replicates (n-5,8) and find that just 21 proteins are slightly differentially expressed.

Because mass spectroscopy is a more sensitive method then western blotting it is not too surprising that the western blot analysis shows no obvious increase in protein amount. I would suggest the authors could evaluate the mRNA levels of the upregulated proteins, because qPCR methods should be sensitive to small changes in transcript levels. This could further support the MS data.

I think the authors do a good job in accurately describing their findings and describing the limitations their study has. I recommend publication after addressing a few very minor concerns and if deemed important include analysis on the transcriptional level.

Minor concerns:

Figure 3 A - The western blot is lacking a loading control to compare how much sample was run between control and CRVO model. The changes should be normalized to a housekeeping protein loading control. The label on the y-axis is small and difficult to read.

Figure 6 C,D the axis labels are too small and hard to read.

Author Response

In this manuscript the authors study the changes of protein levels using tandem mass tag based mass spectrometry after the administration of the drug aflibercept in pigs eyes that were healthy or were suffering from central retinal vein occlusion. The paper is written in clear language and the methods include the important details to allow for reproducibility of the study. The introduction is on point in providing enough details to put conducted study into context, without being too meandering. The authors perform the analysis in a satisfying amount of replicates (n-5,8) and find that just 21proteins are slightly differentially expressed.

RESPONSE: Thank you for your time and comments.

Because mass spectroscopy is a more sensitive method then western blotting it is not too surprising that the western blot analysis shows no obvious increase in protein amount. I would suggest the authors could evaluate the mRNA levels of the upregulated proteins, because qPCR methods should be sensitive to small changes in transcript levels. This could further support the MS data.

RESPONSE: We agree with the reviewer that qPCR is another important strategy. However, our experiments were not originally designed for qPCR analysis. Running qPCR on our samples would have required a different approach by the time of sample collection to ensure that the tissue would be suitable for
qPCR. Performing qPCR would require us to conduct new animal experiments on another five animals. Regulations from Danish Animals Experiments Inspectorate require us to use the minimal number of animals.

I think the authors do a good job in accurately describing their findings and describing the limitations their study has. I recommend publication after addressing a few very minor concerns and if deemed important include analysis on the transcriptional level.

RESPONSE: Thank you for your kind remark.

Minor concerns:
Figure 3 A - The western blot is lacking a loading control to compare how much sample was run between control and CRVO model. The changes should be normalized to a housekeeping protein loading control. The label on the y-axis is small and difficult to read.

RESPONSE: A loading control was added in the revised manuscript. The Western blot for galectin-3 was performed with beta-tubulin as a housekeeping protein loading control to which the changes were normalized. In the revised manuscript, we emphasized that beta-tubulin was used as a loading control.

Lines: 112-113

Figure 6 C,D the axis labels are too small and hard to read.

RESPONSE: The axis labels of Figures 6C and 6D were enlarged as requested.

Lines: 173-174

Reviewer 2 Report

This manuscript describes the effect of Aflibercept intervention in experimental central retinal occlusion. The title is about proteome analysis, however, I found only a minimal data on proteomics, and significant part of the manuscript deals with model evaluation and and immunohistochemistry. Majority of the 21 reported changes in proteins following Aflibercept intervention show minimal changes ranging 0.79 to 1.19 except 3 proteins (Table 1). The p-values of these proteins are barely significant (close to 0..05), which makes it harder to think that these proteins are indeed statistically significant and can be linked to biology. Without rigorous statistical analysis like q-value calculation, it would be hard to convince reader that these are indeed significant changes. Even if that is the case, proteomics data are so minimal that I don't think this is sufficient to publish this manuscript as a proteomics paper.

Moreover, the proteomics experimental section is poorly described, so its hard to evaluate exactly how rigorous proteomic experiments were. Authors should provide details of the experimental workflow for proteomics, detail statistical analysis.   

Author Response

This manuscript describes the effect of Aflibercept intervention in experimental central retinal occlusion. The title is about proteome analysis, however, I found only a minimal data on proteomics, and significant part of the manuscript deals with model evaluation and immunohistochemistry.

Thank you for your review of our manuscript and for your time. In our university hospital, we give more than 50 injections daily. As aflibercept is a very frequently used therapeutic agent, we found it essential to verify the reproducibility of the CRVO model prior to testing aflibercept. Based on our previous proteome studies of the retina, we learned that proteomic analysis of the retina differs significantly from other ocular tissues due to the complex multilayered structure of the retina. We have previously shown that proteome changes of the
retina often occur at very specific layers of the retina (Cehofski et al. Exp Eye Res. 2018 Jun;171:174-182). Therefore, we find it necessary to confirm proteome changes with immunohistochemistry to show the specific layers where the proteome changes occur.

Majority of the 21 reported changes in proteins following Aflibercept intervention show minimal changes ranging 0.79 to 1.19 except 3 proteins (Table 1). The p-values of these proteins are barely significant (close to 0..05), which makes it harder to think that these proteins are indeed statistically significant and can be linked to biology. Without rigorous statistical analysis like q-value calculation, it would be hard to convince reader that these are indeed significant changes. Even if that is the case, proteomics data are so minimal that I don't think this is sufficient to publish this manuscript as a proteomics paper.

Thank you for your comment. We agree that proteome changes following aflibercept intervention were modest. As proteome studies of aflibercept intervention in the experimental CRVO model have not been
performed previously, we could not possibly predict the effect of aflibercept on the retinal proteome. Although aflibercept was found to cause slight changes in a limited number of proteins, we find that our results are worth reporting as our findings have significant consequences for our patients. As stated, aflibercept was only found to cause slight changes in a limited number of proteins. This is an important observation for a number of reasons. Firstly, the results suggest that aflibercept does not regulate a multitude of proteins and signaling pathways. This is reassuring for our patients as they receive a therapeutic agent that works on the VEGF signaling pathway without regulating a multitude of other pathways that could have unwanted side effects. Secondly, as clinicians we find the observations useful in our decision-making. The results of this proteomics study indicate that aflibercept has a narrow mechanism of action in CRVO. In cases where aflibercept does not have any major effect on clinical end-points, clinicians should consider switching treatment due to the narrow mechanism of action observed for aflibercept.

Moreover, the proteomics experimental section is poorly described, so its hard to evaluate exactly how rigorous proteomic experiments were. Authors should provide details of the experimental workflow for proteomics, detail statistical analysis.

RESPONSE: Many of the proteomic techniques used for proteomic analysis of the retina were described in detail in our previous studies. We feel it is more correct to direct the reader to previous studies rather than providing a repetition of the methods. 

Reviewer 3 Report

It is not clear to me why CRVO was induced in seven animals and only five chosen to check the reproducibility of proteomic experiments and only two for immunohistochemistry. If not explained, these numbers make me confused.

Author Response

It is not clear to me why CRVO was induced in seven animals and only five chosen to check the reproducibility of proteomic experiments and only two for immunohistochemistry. If not explained, these numbers make me confused.

RESPONSE: Thank you for your time and for your review of our manuscript. We clarified the details in the revised manuscript. Eyes for proteomic analysis and eyes for immunohistochemistry need different handling and preparation. In our experience, it is difficult to collect retinal tissue for proteomics and
immunohistochemistry from the same eye. Therefore, eyes are selected for either proteomics or immunohistochemistry. For example, if retinal tissue is collected for proteomic analysis. For example, collection of retinal tissue for proteomic analysis may render the remaining retinal tissue poorly suited for immunohistochemistry as collection of retinal tissue may cause retinal detachment.

Lines 244-245

Round 2

Reviewer 2 Report

The method section must be improved and requires some details.  Authors cited their previously published paper but I will suggest providing brief addition of information about LC and mass spec conditions, database searches and analysis. This is for convenience of readers and also reviewers.

I also suggest to write a small section at the end about the limitation of proteomic analysis. It can be added at the end of the conclusion.   

Reviewer 3 Report

The article may be accepted for publication in its present form.
